# Warm and Harsh Parenting, Self-Kindness and Self-Judgment, and Well-Being: An Examination of Developmental Differences in a Large Sample of Adolescents

**DOI:** 10.3390/children10020406

**Published:** 2023-02-19

**Authors:** Yizhen Ren, Xinli Chi, He Bu, Liuyue Huang, Shaofan Wang, Ying Zhang, Di Zeng, Hao Shan, Can Jiao

**Affiliations:** 1Faculty of Psychology, Beijing Normal University, Beijing 100875, China; 2School of Psychology, Shenzhen University, Shenzhen 518061, China; 3The Shenzhen Humanities & Social Sciences Key Research Bases of the Center for Mental Health, Shenzhen 518061, China; 4Department of Social and Behavioral Sciences, City University of Hong Kong, Tat Chee Avenue, Kowloon, Hong Kong SAR, China; 5Department of Psychology, Faculty of Social Sciences, University of Macau, Macao SAR, China; 6Center for Cognitive and Brain Sciences, Institute of Collaborative Innovation, University of Macau, Macao SAR, China; 7Faculty of Education, Beijing Normal University, Beijing 100875, China

**Keywords:** warm parenting, harsh parenting, self-kindness, self-judgment, adolescent well-being

## Abstract

*Objectives:* This study aimed to examine the associations between warm and harsh parenting and adolescent well-being, and the mediating effects of self-kindness and self-judgment, in relationships. Moreover, this study investigated developmental differences across three adolescence stages (early, middle, and late). *Methods:* In total, 14,776 Chinese adolescents (mean age = 13.53 ± 2.08, 52.3% males), including individuals in early (10–12 years old, *N* = 5055), middle (13–15 years old, *N* = 6714), and late adolescence (16–18 years old, *N* = 3007) participated in this study. All the adolescents rated their levels of warm and harsh parenting, self-kindness and self-judgment, and well-being. Structural equation modeling (SEM) was adopted to examine the mediation model. Multi-group analysis was conducted to investigate differences in the mediation model across the different developmental stages. *Results:* Both warm and harsh parenting were related to adolescent well-being through the mediating effects of self-kindness and self-judgment. However, warm parenting exerted a more substantial impact on adolescent well-being. Self-kindness had a more robust mediating effect than self-judgment in relationships. Moreover, harsh parenting had a weaker impact on adolescent well-being in late adolescence than in early and middle adolescence. Warm parenting had a more significant impact on adolescent well-being in early adolescence than in middle and late adolescence. *Conclusions:* Overall, warm parenting had a more substantial effect than harsh parenting on adolescent well-being. The findings also highlighted the crucial mediating effect of self-kindness in the relationships between parenting and well-being. Moreover, this study also indicated the importance of warm parenting in early adolescence. Intervention programs should focus on enhancing the level of warm parenting to promote self-kindness in adolescents, in order to improve their well-being.

## 1. Introduction

Well-being is recognized as critical in adolescent development. Based on the ecological perspective of adolescent development [1,2,3], family and parental factors play significant roles in adolescent well-being [4,5,6,7]. Parenting style, as demonstrated by parents’ emotional attitudes toward the child [8], affects adolescent well-being [9] by influencing the ways adolescents connect to themselves [10]. The current study investigated whether warm and harsh parenting, as two typical representations of parenting style dimensions [11,12], would relate to adolescent well-being through the mediating effects of self-kindness and self-judgment. Due to the nature of turbulence in adolescence [13], the associations may differ across early, middle, and late adolescence stages [10], which have yet to be fully addressed in the existing literature. Therefore, this study further investigated whether there were developmental differences in the relationships across the adolescence stages.

### 1.1. Parenting and Adolescent Well-Being

According to ecological systems theory [1,2,3], adolescent development is shaped by interactions between individuals and their contexts (e.g., family, school, community, and society). The family has been acknowledged as the primary proximal socialization environment that exerts an immediate on adolescents [5,6,10,14]. Accordingly, adolescent adjustments are embedded in children’s proximal interaction processes with parents within the family [1,3], which has also been emphasized by family system theory [15,16]. Parenting behavior, constituting the most proximal family interaction environment [3,17], has been shown to be important in promoting adolescent adjustments [17,18,19,20]. The present study explores the relationship between parenting behavior and adolescent well-being, with a particular emphasis on the significance of parenting style within the family setting.

According to the integrative model of parenting [8], parenting style, referring to the emotional attitude of parents toward the child, plays a crucial role in shaping a good parent–child relationship and children’s adjustments [7,18]. Parenting styles have been classically conceptualized as two broad parenting dimensions: responsiveness and demandingness [11]. Responsiveness refers to parents’ providing of support, responsive care, and sensitivity to children’s needs to aid their development, while demandingness refers to their supervising and enforcing discipline efforts [11,12]. The current study focused on warm and harsh parenting as the representative parenting styles for responsiveness and demandingness. Warm and harsh parenting are crucial in forming and maintaining a good parent–child relationship and have significant impacts on children’s multiple psychological adjustment outcomes [9,21,22,23,24]. However, few studies have simultaneously analyzed both warm and harsh parenting styles to determine their relative impact on adolescent well-being. Well-being refers to subjective feelings of happiness and enjoyment as well as healthy and prosperous individual functioning [10,14]. However, previous studies [4,5,6] assessing adolescent well-being have mainly used other indicators, such as life satisfaction and helplessness, rather than adolescent well-being itself. Moreover, with particular reference to adolescents, the well-being of this group is highly susceptible to external factors during the crucial developmental period [5,6,7,10,14]. Hence, it is meaningful to investigate the relative importance of warm and harsh parenting for adolescent well-being.

### 1.2. The Mediating Roles of Self-Kindness and Self-Judgment

To improve intervention programs, the mechanisms connecting parenting and adolescent well-being need clarification. Attachment theory suggests that parenting styles (warm and harsh) shape children’s internalized emotional attitudes, which influence the way children connect to themselves [9,10,25]. Therefore, the current study focused on self-kindness and self-judgment as potential mediators in the relationships. Self-kindness has been described as a beneficial way of relating to oneself, while self-judgment reflects harshness and strictness with oneself [26]. In Gilbert’s social mentality theory, self-kindness can be considered as a kind of self-soothing behavior in the safeness system, which can downregulate the activated threat system, leading to better psychological outcomes [26,27]. However, self-judgment may stimulate the threat–defense system, elevating stress responses [26,27], and thus finally leading to decreasing levels of well-being.

Attachment theory could help us understand the mediating effects of self-kindness and self-judgment [9,10,25]. According to attachment theory, individuals gradually develop internal representations of the self and others from interactions with important caregivers, especially parents [25]. The attachment representations indicate not only expectations about the availability of other important figures but also feelings of self-worth and self-evaluation [10,28,29]. Adolescents are likely to internalize the emotional attitudes of their parents and formulate their attachment representations [10,23]. For example, if parents adopt harsh parenting toward adolescents and demonstrate intolerance toward their mistakes, the adolescents may formulate the beliefs that they are “bad” or unworthy of love, and become less self-kind, demonstrating more self-judgment instead [10], which may influence their well-being.

Moreover, according to parental acceptance–rejection theory [30,31], children’s psychological adjustment can be predicted by their perceptions of the extent of their parents’ acceptance or rejection of them. Children with perceptions of acceptance from parents have better psychological outcomes, such as greater emotional functioning and healthier personality dispositions [30,31]. Parental warmth can make children feel that they are being accepted and loved unconditionally [22,23], as well promoting trust and reciprocity in the parent–child relationship [23], promoting their self-esteem levels [24] and facilitating the children’s well-being. However, harsh parenting may make children perceive emotional rejection from parents, formulate negative evaluations about themselves, and finally experience devastating consequences such as hostility, negative mood states, and emotional instability [30,31,32,33]. Children who receive signals of rejection from parents tend to make negative evaluations of themselves and develop low levels of self-esteem, self-worth, and self-efficacy [30,31,32,33], and are more likely to have self-judgment, which may devastate their well-being.

### 1.3. Developmental Differences

Warm and harsh parenting may affect adolescent well-being differently in different developmental stages (i.e., early, middle, and late adolescence) because adolescents undergo rapid and profound changes across adolescence. The differences may exist at both variable and relationship levels. In terms of the variable level, adolescence is a critical period for decreasing levels of well-being [7] and life satisfaction [6], and instead leads to more and more hopelessness [6,7]. Moreover, previous research has also suggested that gradual declines in negative parenting behaviors enhance parent–child relationship quality [7]. In terms of the relationship level, a previous study found that the impact of parental psychological control on adolescents’ meaning of life was more significant in early adolescence than in late adolescence [5]. Researchers have attributed this to adolescents’ growing autonomy and competence, making them less susceptible to parenting behaviors as they get older [5]. However, no previous studies have investigated the developmental differences in relationships among warm and harsh parenting, self-kindness, self-judgment, and adolescent well-being.

### 1.4. The Current Study

This study examined the relationships and mediating mechanisms between warm and harsh parenting and well-being among a large sample of Chinese adolescents. Moreover, this study investigated the developmental differences across early, middle, and late adolescence. We hypothesized that warm and harsh parenting would relate to well-being through increased self-kindness and decreased self-judgment among adolescents. Moreover, we hypothesized that the relationships may differ across the different adolescence stages, such that warm and harsh parenting may have a greater impact on well-being in the early adolescence stage (please Figure 1).

## 2. Methods

### 2.1. Procedure

In December 2021, we collected the data for this study in Shenzhen, Guangdong Province, China. With the support of the Shenzhen Academy of Educational Sciences, a convenience sampling method was used to collect data in 28 primary and secondary schools. The Human Research Ethics Committee of the corresponding authors’ university approved the methods generating the data used in the study (Code number: PN-2021-0069). Teachers informed participants about the study before data collection. Participants completed online questionnaires in the school computer room upon giving consent. They were assured that all personal information would be kept strictly confidential.

### 2.2. Participants

The participants included 15,492 students, of whom 14,776 (95.38%) returned valid questionnaires. A total of 7728 boys and 7048 girls were enrolled in the current study. The participants’ mean age was 13.53 (SD = 2.08). Adolescents belong to three developmental stages [13,34], including early (10–12 years old, *N* = 5055), middle (13–15 years old, *N* = 6714), and late adolescence (16–18 years old, *N* = 3007). In the sample, 25.6% of the participants were the only children in their families. In the total model, age, gender, and only-child status were treated as control variables in the analysis (see Table 1).

### 2.3. Measurements

#### 2.3.1. Warm and Harsh Parenting

Warm and harsh parenting were assessed by two subscales of the brief version of the Alabama Parenting Questionnaire (APQ) [35]. The warm parenting subscale includes three items (i.e., “My parents praise me if I behave well”, “My parents let me know when I am doing a good job with something”, and “My parents compliment me when I do something well”). The harsh parenting subscale also includes three items (i.e., “My parents spank me with their hands when I have done something wrong”, “My parents slap me when I have done something wrong”, and “My parents hit me with a belt, switch, or other objects when I have done something wrong”). Each item is rated on a Likert scale of 1 (never) to 5 (always). The Chinese version of the APQ has been validated and widely used in Chinese populations [35,36]. In the current study, the Cronbach’s α of warm and harsh parenting reported by adolescents were 0.94 and 0.85, respectively.

#### 2.3.2. Self-Kindness and Self-Judgment

Adolescent self-kindness and self-judgment were assessed using the self-compassion scale—youth version (SCS–Y), which has demonstrated good reliability and validity in Chinese adolescents [37,38]. It consists of 17 items divided into six subscales: self-kindness, self-judgment, common humanity, isolation, mindfulness, and over-identification. Each item is rated on a five-point Likert scale from 1 (almost never) to 5 (almost always). The items for self-kindness and self-judgment were used in the current study. Higher scores indicate higher levels of self-kindness and self-judgment. The Cronbach’s α of self-kindness and self-judgment in the present study were 0.80 and 0.70, respectively.

#### 2.3.3. Well-Being

Adolescent well-being was assessed by the WHO-Five Well-Being Index [39,40], which was developed by the World Health Organization. Each item is rated on a six-point scale ranging from 0 (all of the time) to 5 (at no time). Higher scores indicate higher levels of well-being. Good reliability and validity have been reported in Chinese populations [41]. The Cronbach’s α in this study was 0.95.

#### 2.3.4. Control Variables

In the children’s questionnaires, we collected the child’s age, gender (male or female), and only-child status (i.e., whether their family had only one child or more than one child). All of these variables were controlled in the subsequent analysis.

### 2.4. Data Analysis

Descriptive statistics and Pearson correlational analysis were conducted in SPSS 23.0. The present study performed structural equation modeling (SEM) to examine the mediation model in Mplus 8.3. Bootstrap analysis with 2000 replicates and a 95% confidence interval was used to examine the significance of the mediation effects. Adolescent characteristics, including age, gender, and only-child status were treated as control variables in the analysis. The standardized summed composite scores of each scale were used in the SEM analysis. Then we conducted multi-group analysis to investigate differences in the mediation model across different developmental stages (early adolescence: 10–18 years old; middle adolescence: 13–15 years old; and late adolescence: 16–18 years old).

## 3. Results

### 3.1. Preliminary Analysis

Table 2 presents the study variables’ means, standard deviations, and Pearson correlations. This section may be divided by subheadings. It should provide a concise and precise description of the experimental results and their interpretation, as well as the experimental conclusions that can be drawn.

### 3.2. The Mediation Model

After controlling for a series of confounding variables, the tested model obtained acceptable fit indices (RMSEA = 0.070, CFI = 0.961, TLI = 0.952, SRMR = 0.042). The details of the mediation model are shown in Table 3 and the specific paths of relationships appear in Figure 2. As Table 3 illustrates, the 95% bootstrap confidence interval of the mediation effects of self-kindness and self-judgment did not include zero. Thus, these mediation effects were all statistically significant. In the overall model, warm and harsh parenting were also directly related to adolescent well-being (β = 0.199, β = −0.073, *p* < 0.001). The bootstrap tests showed that warm parenting had a greater impact than harsh parenting on adolescent well-being (95%CI [0.229, 0.280]).

We also conducted bootstrap analysis to examine the mediating effect differences (the mediating effect of self-kindness versus the mediating effect of self-judgment). The results showed that the mediating effect of self-kindness was more robust than that of self-judgment in the relationship between warm parenting and adolescent well-being (95%CI [0.126, 0.147]). In the relationship between harsh parenting and adolescent well-being, self-kindness also demonstrated a stronger mediating effect than self-judgment (95%CI [−0.022, −0.007]).

### 3.3. Multi-Group Analysis across Developmental Stages of Adolescence

In the multi-group analysis, the metric invariance and scalar invariance of the measurement model across adolescent developmental stages were verified. The model with free estimated structural paths fitted data well (*χ*^2^ = 2076.899, *df* = 117, RMSEA = 0.058, CFI = 0.976, TLI = 0.970, SRMR = 0.025). The model with constrained estimated structural paths also fitted the data well (*χ*^2^ = 2178.556, *df* =133, RMSEA = 0.056, CFI = 0.975, TLI = 0.972, SRMR = 0.030). The differences between the model with free estimated paths and the model with constrained paths reached significant levels (Δ*χ*^2^ = 101.657, Δ*df* = 16, *p* < 0.001). This indicated developmental differences across the three adolescence stages.

## 4. Discussion

The current study constructed a mediation model in which warm and harsh parenting were related to adolescent well-being through self-kindness and self-judgment. This study revealed that both warm and harsh parenting could influence adolescent well-being through the mediating effects of self-kindness and self-judgment. However, warm parenting exerted a more substantial impact than harsh parenting on adolescent well-being. Moreover, self-kindness demonstrated a more robust mediating effect than self-judgment in the relationships. In addition, warm parenting exerted a greater impact on adolescent well-being in early adolescence than in middle and late adolescence. Harsh parenting in early and middle adolescence had a greater impact on adolescent well-being than was the case in late adolescence. Notably, even though the effect of warm parenting was a little weaker in middle and late adolescence than in early adolescence, it was still significant across all developmental stages, thereby highlighting the importance of warm parenting for adolescent well-being.

### 4.1. Warm and Harsh Parenting and Adolescent Well-Being

The current study found that warm and harsh parenting were related to adolescent well-being through the mediating effects of self-kindness and self-judgment. Moreover, warm parenting exerted a more substantial impact on adolescent well-being. Warm parenting could convey parents’ love and care to children, fulfill the basic psychological demands of children [23,42,43,44], and improve their well-being. Through warm parenting, parents can establish a warm emotional climate in the family [23], promote a good parent–child relationship [23], and improve emotional communications, all of which can improve adolescents’ well-being [44]. However, harsh parenting may lead to emotional fluctuations, hinder the parent–child relationship [45], and negatively influence adolescent well-being. Moreover, harsh parenting may be developmentally inappropriate for adolescents as they tend to obtain more and more independence and autonomy [12,46]. Under harsh parenting, adolescents may gradually lose confidence in their academic performance, have more negative perceptions of their competence in multiple domains [47], develop a pessimistic attitude toward the world and themselves, and finally experience a low level of well-being.

### 4.2. The Mediating Effects of Self-Kindness and Self-Judgment

The current study found significant mediating effects of self-kindness and self-judgment in the relationships. However, most notably it showed that it was mainly through increasing or decreasing self-kindness that parenting impacted adolescent well-being. This is similar to some previous results demonstrating a stronger connection between self-kindness and positive psychological indicators than between self-judgment and positive psychological indicators [48,49]. In a previous study [49], researchers identified the significant effects of self-kindness on children’s positive personal growth. However, they found that self-judgment could not significantly influence adolescents’ personal growth [49]. Another study also revealed stronger associations between self-kindness and positive psychological indicators, such as the meaning of life and life satisfaction, than was the case for self-judgment [48]. In contrast, self-judgment demonstrated stronger relationships with negative psychological adjustments [50,51]. Therefore, this study highlights the importance of enhancing self-kindness in adolescents to facilitate their well-being.

According to attachment theory [9,10,25] and parental acceptance–rejection theory [30,31], warm parenting can promote more positive interactions in the family [23], which may help children develop self-compassion skills [10]. Researchers have proposed that individuals can better develop self-soothing skills within security-boosting interactions with significant attachment figures [23,27,52]. Warm parenting can provide more caring, love, and sensitive responsiveness, and improve the attachment relationships between parents and the child [23], thereby creating the necessary foundations for a soothing system and the development of self-kindness [26], finally improving their well-being. In contrast, harsh parenting of adolescents does not provide the fundamental conditions for developing secure attachment and, consequently, self-compassion [10]. In such an emotional context, the soothing system is insufficiently developed, and the child is more likely to be self-judgmental rather than self-kind [26], leading to decreasing well-being levels.

### 4.3. Developmental Differences

In the current study, it was shown that warm parenting could significantly improve adolescent well-being across early, middle, and late adolescence, thereby highlighting the importance of warm parenting for adolescents. However, there were still differences. Specifically, the total effect of warm parenting was higher in early adolescence than in middle and late adolescence (See Figure 3, Figure 4 and Figure 5). This may be because during early and middle adolescence the development from early to middle adolescence parallels the transition from elementary to middle school [12,13]. During this period, adolescents experience dramatic changes [12,47] and may be more emotionally susceptible to family influences. Moreover, previous studies also revealed the significance of parental behaviors, especially parental responses to adolescent emotions, during early adolescence [47,53].

The total effect of harsh parenting on adolescent well-being in early adolescence was similar to that seen in middle adolescence. However, the effect in late adolescence was much weaker. Moreover, harsh parenting was not significantly related to self-kindness and not directly related to adolescent well-being in late adolescence (See Figure 5). The decreasing susceptibility of adolescent well-being to harsh parenting can be attributed to the presence of greater developmental autonomy in late adolescence [5,13]. As adolescents grow older, they may become more independent [13] and are less likely to be affected by others.

In terms of the results, we found that perceived parenting style exerted stronger impacts on adolescent well-being during early adolescence than in late adolescence. The results were similar to previous findings [5,54], indicating different patterns of parenting influence across adolescence stages. For example, parental psychological control have different effects on the meaning of life in the early and late adolescence stages [5], which is possibly due to adolescents’ growing autonomy and independence and reduced susceptibility to parenting. From another point of view, this may also be partially explained by developmental changes in parenting [55], such as decreases in warmth and behavioral control as children age from 8 to 16, with sharper declines particularly after early adolescence.

### 4.4. Future Directions

This study did not examine possible gender differences. Previous studies have suggested that the child’s gender would influence perceptions of parenting style, adolescent self-attitudes [34,35,36], and psychological adjustments [31,32,33]. For example, one study [7] found a stronger association between parental psychological control and adolescent well-being in adolescent girls than in boys. Particularly for girls, harsh parenting would adversely influence adolescents’ sense of meaning in life and self-control, which enhances their problematic smartphone use, [56]. Additionally, harsh parenting was only directly and indirectly related to female depression, but not to male depression [57], highlighting the specific importance of harsh parenting for girls. Researchers [58,59,60] have also revealed potential gender differences in terms of self-attitudes. For example, self-identified men tended to have more self-compassion than women toward themselves [58]. Older females had the lowest self-compassion levels, compared with younger females or all-age males [59]. Future research may adopt multiple methods and measures to examine possible age and gender differences in these associations.

In our study, we found that harsh parenting perceived by adolescents had a weaker impact than warm parenting on their well-being. This may be partially explained by the Chinese cultural context. Chinese culture, for example, attaches great importance to collectivism and filial piety, which is reflected in traditional values such as respect for elders, obedience to norms, and fulfillment of family obligations [12,61]. Compared to Western parents, Chinese parents are less likely to express love outwardly through physical affection or verbal affirmations [62], are more likely to adopt harsh discipline methods [12], and attach less importance to autonomy [63]. Additionally, Chinese parents often adhere to the traditional belief that they should be both strict and affectionate toward their children [12]. For example, a previous study [46] identified four types of parenting styles in Chinese American parents, including “supportive”, “easygoing”, “harsh,” and “tiger” parenting. Notably, a significant proportion of Chinese families were found to adopt harsh or “tiger” parenting [46]. Under this circumstance, it is possible that adolescents in China may view strict or harsh parenting as a normal and expected part of family life. As a result, the negative effects of this type of parenting on adolescent well-being may be less pronounced. Furthermore, our study did not distinguish between fathers’ and mothers’ parenting roles. Considering the traditional image of parents, “strict father, kind mother” within families [64], parental gender may also play an important role in understanding the influence of harsh parenting on adolescent well-being. Future research may distinguish between fathers’ and mothers’ warm and harsh parenting to investigate the specific associations with adolescent well-being.

Finally, this study was conducted during the pandemic in China. Empirical studies have indicated the adverse influence of the pandemic on families [65,66,67]. During the pandemic, parents faced considerable stress in balancing multiple demands among work, life, and family [68]. Parents experienced elevated levels of parenting stress [67], which may have increased the possibility of adopting more harsh parenting [45]. Additionally, a recent study [65] reported a significant deterioration in parenting quality during the pandemic’s initial months compared with the pre-pandemic period. Hence, researchers [65] have stressed the importance of intervention programs to prevent prolonged individual and family problems by offering more support and guidance to families.

Our findings contribute to understanding parenting and well-being in adolescents from the ecological perspective of development. By investigating the mediating mechanism of self-attitudes (self-kindness and self-judgment), our findings suggest that it is important to improve self-kindness to promote adolescent well-being. Moreover, our study indicates the relative importance of warm parenting for adolescent well-being, particularly during early adolescence. Overall, our findings have indicated the significance of improving self-kindness in adolescents, particularly when adolescents experience reduced warm and increased harsh parenting within families.

### 4.5. Limitations

Some limitations need to be noted. First, the cross-sectional research design precluded us from exploring the causal links between parenting and adolescent well-being. Future research should adopt a longitudinal design to explore the casual associations and developmental trends between parenting and adolescent well-being Second, we collected data about parenting based only on adolescents’ reports. Future research might collect multiple sources of data (e.g., parents’ and adolescents’ mutual report) to test the relationships. Third, we only xplored the mediating effects of individual adolescent characteristics in the relationships between parenting and adolescent well-being. Future research might include both relational (i.e., parent–child attachment) and individual adolescent factors simultaneously to investigate the influence on adolescent well-being.

### 4.6. Implications

This study has several implications. First, programs aimed at enhancing warm parenting and reducing harsh parenting may constitute important intervention directions to improve well-being among adolescents. Therefore, it may be particularly beneficial to improve warm parenting for adolescents. Second, it is suggested that self-kindness should be recognized as promoting well-being among adolescents. Parents, teachers, and educators could facilitate the practice of self-kindness to help adolescent improve their well-being. Third, intervention programs should pay attention to improving self-kindness, particularly during early adolescence.

## 5. Conclusions

In conclusion, both warm and harsh parenting were found to be related to adolescent well-being through the mediating effects of self-kindness and self-judgment. However, warm parenting exerted a more substantial impact than harsh parenting on adolescent well-being. Self-kindness was shown to have a more robust mediating effect than self-judgment in the relationships. Moreover, differences between developmental stages were identified in the relationships. Harsh parenting in late adolescence had a weaker impact on adolescent well-being than was the case in early and middle adolescence. Warm parenting exerted a greater impact on adolescent well-being in early adolescence than in middle and late adolescence.

## Figures and Tables

**Figure 1 children-10-00406-f001:**
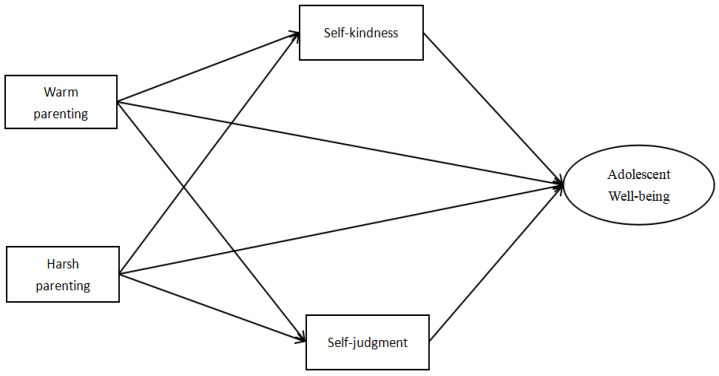
The hypothesized mediation model.

**Figure 2 children-10-00406-f002:**
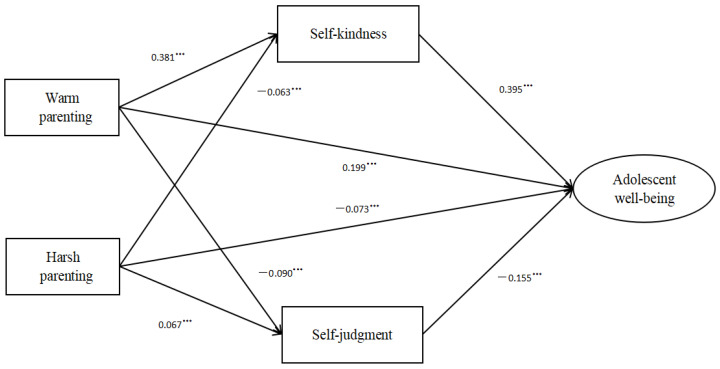
The mediation model in the total sample. *** *p* < 0.001.

**Figure 3 children-10-00406-f003:**
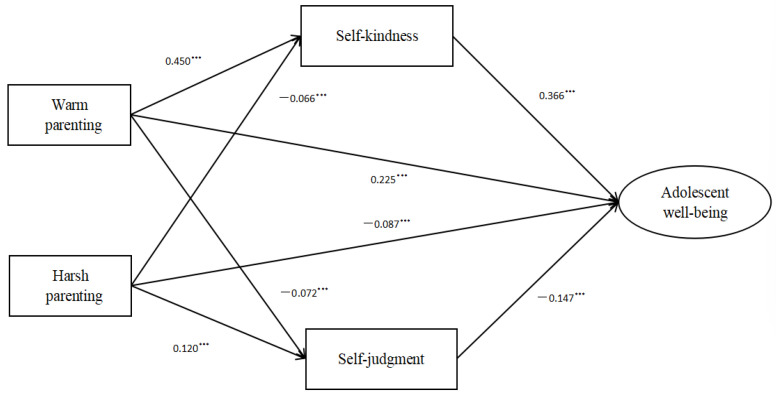
The mediation model in early adolescence. *** *p* < 0.001.

**Figure 4 children-10-00406-f004:**
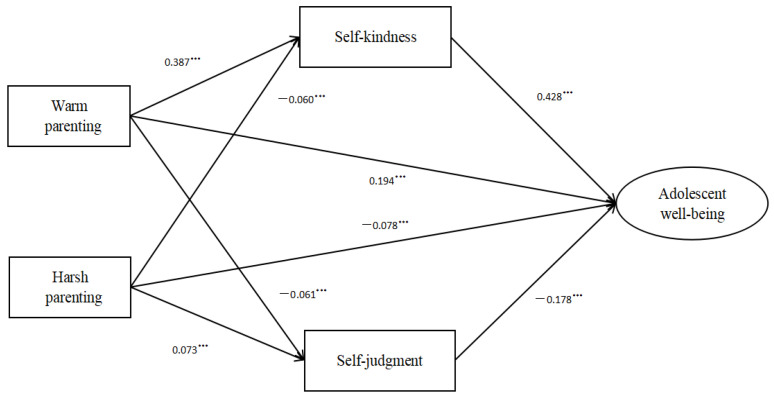
The mediation model in middle adolescence. *** *p* < 0.001.

**Figure 5 children-10-00406-f005:**
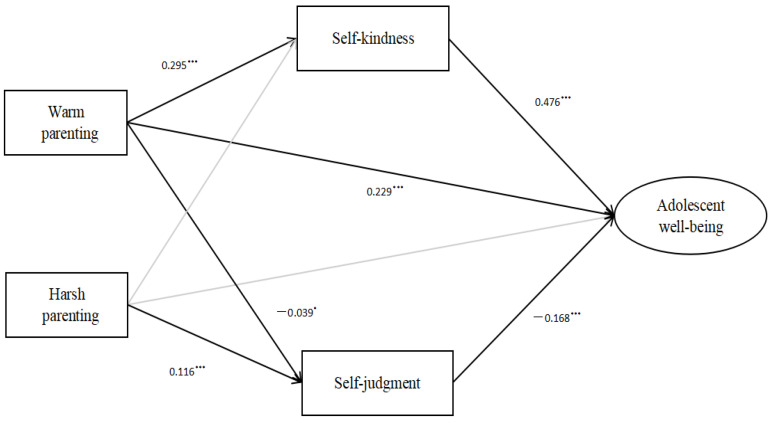
The mediation model in late adolescence. * *p* < 0.05; *** *p* < 0.001.

**Table 1 children-10-00406-t001:** Demographic information of participants.

Variables	Adolescents	
	N/M	%/SD
**Age**	13.53	2.08
**Gender**		
Boys	7728	52.3%
Girls	7048	47.7%
**Only-child status**		
Only child	3777	25.6%
Not an only child	10,999	74.4%
**Educational status**		
Primary school	3098	21.0%
Lower general secondary school	7228	49.0%
Higher general secondary school	4450	30.0%

**Table 2 children-10-00406-t002:** Intercorrelations, means, and standard deviations for study variables.

Variable	1	2	3	4	5	6	7	8
1 gender	1							
2 age	0.044 **	1						
3 siblings	0.007	0.079	1					
4 WAP	−0.129 **	−0.044 **	−0.060 **	1				
5 HAP	−0.140 **	−0.064 **	0.043 **	−0.249 **	1			
6 self-kindness	−0.027 **	−0.054 **	−0.053 **	0.396 **	−0.157 **	1		
7 self-judgment	0.197 **	0.076 **	0.033 **	−0.106 **	0.089 **	−0.049 **	1	
8 adolescent well-being	−0.176 **	−0.096 **	−0.014	0.432 **	−0.192 **	0.529 **	−0.238 **	1
M				8.781	10.279	9.533	7.911	12.906
SD				2.743	2.899	2.623	2.550	4.575

Note. ** *p* < 0.01; WAP = Warm parenting, HAP = Harsh parenting.

**Table 3 children-10-00406-t003:** The bootstrap confidence interval and effect size of the mediation model in total samples.

	Mediation Paths	Estimate	*p*	95% CI
Total sample	WAP→Self-kindness→WEL	0.150	<0.001	[0.141, 0.159]
	WAP→Self-judgement→WEL	0.014	<0.001	[0.011, 0.017]
	HAP→Self-kindness→WEL	−0.025	<0.001	[−0.032, −0.018]
	HAP→Self-judgement→WEL	−0.010	<0.001	[−0.013, −0.007]
	Total effect of WAP (direct plus indirect)	0.363	<0.001	[0.347, 0.379]
	Total effect of HAP (direct plus indirect)	−0.109	<0.001	[−0.124, −0.093]
Early adolescence	WAP→Self-kindness→WEL	0.164	<0.001	[0.148, 0.181]
	WAP→Self-judgement→WEL	0.011	<0.001	[0.005, 0.016]
	HAP→Self-kindness→WEL	−0.024	<0.001	[−0.035, −0.013]
	HAP→Self-judgement→WEL	−0.018	<0.001	[−0.023, −0.012]
	Total effect of WAP (direct plus indirect)	0.400	<0.001	[0.370, 0.431]
	Total effect of HAP (direct plus indirect)	−0.129	<0.001	[−0.154, −0.104]
Middle adolescence	WAP→Self-kindness→WEL	0.165	<0.001	[0.151, 0.179]
	WAP→Self-judgement→WEL	0.011	<0.001	[0.006, 0.016]
	HAP→Self-kindness→WEL	−0.025	<0.001	[−0.037, −0.014]
	HAP→Self-judgement→WEL	−0.013	<0.001	[−0.018, −0.008]
	Total effect of WAP (direct plus indirect)	0.370	<0.001	[0.345, 0.395]
	Total effect of HAP (direct plus indirect)	−0.117	<0.001	[−0.142, −0.092]
Late adolescence	WAP→Self-kindness→WEL	0.141	<0.001	[0.121, 0.160]
	WAP→Self-judgement→WEL	0.006	0.046	[<0.001, 0.013]
	HAP→Self-kindness→WEL	−0.016	0.091	[−0.035, 0.003]
	HAP→Self-judgement→WEL	−0.019	<0.001	[−0.028, −0.011]
	Total effect of WAP (direct plus indirect)	0.376	<0.001	[0.344, 0.409]
	Total effect of HAP (direct plus indirect)	−0.063	0.002	[−0.102, −0.023]

Note. WAP = Warm parenting, HAP = Harsh parenting, WEL = Adolescent well-being.

## Data Availability

The datasets generated during and/or analyzed during the current study are available from the corresponding author on reasonable request.

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
