# Peer review of "Warm and Harsh Parenting, Self-Kindness and Self-Judgment, and Well-Being: An Examination of Developmental Differences in a Large Sample of Adolescents"

_children, 2023, doi:10.3390/children10020406_

Round 1

Reviewer 1 Report

Some strong points of the article can be highlighted.

The number of study participants is very large, forming a representative sample.

The applied research tool is recently validated.

The discussions are elaborated in close connection with the research results and are correlated with other studies.

The topic of the article is of current interest.

Authors should pay attention to several aspects as follows:

1. The technical editing criteria of the article are not respected.

2. The research hypotheses are clearly defined, but their numbering is not logical.

3. Bibliographic references must be corrected according to the style of the journal.

Author Response

Title of paper: Warm and harsh parenting, self-kindness and self-judgment, well-being: An examination of developmental differences in a large sample of adolescents

Dear reviewers,

Thank you very much for your review of our manuscript. We greatly appreciate all your comments,

and these comments are helpful and constructive in revising this paper.

We have carefully revised the manuscript and addressed your comments. Our responses to each comment are enclosed in the response letter, and the revised parts have been highlighted in the revised manuscript. We hope the responses and revisions to our manuscript are thorough and precise. We look forward to hearing from you at your earliest convenience.

Sincerely Yours,

All the authors.

Reviewer(s)' Comments to Author:
Reviewer: 1

Comments to the Author

The topic of the article is of current interest.

Authors should pay attention to several aspects as follows:

  1. The technical editing criteria of the article are not respected.

Responses: Thanks for your critical comments. We feel very sorry for the incorrect technical editing. We have re-edited the manuscript strictly. Thanks very much for your suggestions.

  1. The research hypotheses are clearly defined, but their numbering is not logical.

Responses: Thank you very much for the comments. We have reorganized our research hypothesis as such (See line 140-147)

The current study

This study examined the relationships and mediating mechanisms between warm and harsh parenting and well-being among a large sample of Chinese adolescents. Moreover, this study investigated the developmental differences across early, middle, and late adolescent stages. We hypothesized that warm and harsh parenting would relate to well-being through increased self-kindness and decreased self-judgment among adolescents. Moreover, we hypothesized that the relationships may differ across different adolescent stages, such that warm and harsh parenting may have a greater impact on well-being in the early adolescence stage.

  1. Bibliographic references must be corrected according to the style of the journal.

Responses: Thanks for your critical comments. We have edited the bibliographic references correctly. We will pay attention to this issue in our future manuscript. Thank you very much!

Reviewer 2 Report

I am enthusiasm for authors’ work. The aim of this manuscript was to examine the relationships among warm and harsh parenting, self-kindness and self-judgment, and adolescent well-being taking into account three different adolescent stages (early, middle and late adolescent). Strengths of the manuscript include the importance of the proposed topic carried out in the Asian population and consider adolescent stage from a different range ages perspective that is adequate and convincing. The study is very interesting and could provide a relevant contribution to the understanding of the significant impacts of parental patterns on adolescent well-being.

---Specific comments---

1. The abstract is insufficiently described. Authors should include in this section the statistical analyzes carried out and specific -not generally- conclusions and/or practical implications.

2. Authors could improve their literature review in the Introduction section about some aspects. First, I think it makes more sense and I recommend starting the introduction with a brief summary of the importance of the family and family patterns and how they are associated with adolescent adjustment. In the current paper this summary is focused on well-being variable. Second, the study is focused on three different adolescence stages (early, middle, and late). For this reason, the entire literature review should clearly reflect on with age range of adolescence the cited studies and not refer to the general adolescent population as the section is written right now. So, I recommend deleting the section entitled "developmental differences" and integrating this information throughout the introduction, since the entire study is carried out taking into account the different age ranges of this developmental period.

3. I recommend delete figure 1. The mediation model should be well explained in the current study section and practically is the same as figure 2, it does not provide new information.

4. The present study needs revision. Please, make sure all objectives and hypotheses are stated in this section and explain the results expected. Moreover, I recommend authors delete all information related to study objectives throughout the introduction because they are repetitive and not friendly reading (e.g., “The current study investigated whether warm and harsh parenting, as two typical representations of parenting style dimensions (Baumrind, 1991; Zhang et al., 2017), could relate to adolescent well-being through the mediating effects of adolescent self-kindness and self-judgment” on page 5, “In this case, the current study examined the relationships between warm and harsh parenting and adolescent well-being” on page 6; “Therefore, the current study focused on self-kindness and self-judgment as potential mediators in relationships” on page 7).

5. I assume that based on the total sample (N = 14,776) the authors created the three groups of adolescents (early, middle, and late). This aspect should be clearly explained in the “Data analysis” section.

6. For all measures used, please indicate whether response scores were summed or averaged to create their composite scores for data analysis.

7. Because of parental patterns are different in function on cultural context, I believe the manuscript would gain value if in the Discussion section including a cultural lens to interpret the results found.

8. Please describe the contributions of the study more clarify and extensity.

9. At the end of the manuscript, the practical applications/implications of the study should be explained in details.

Author Response

Submission ID number: children-2140835

Title of paper: Warm and harsh parenting, self-kindness and self-judgment, well-being: An examination of developmental differences in a large sample of adolescents

Dear reviewers,

Thank you very much for your review of our manuscript. We greatly appreciate all your comments,

and these comments are helpful and constructive in revising this paper.

We have carefully revised the manuscript and addressed your comments. Our responses to each comment are enclosed in the response letter, and the revised parts have been highlighted in the revised manuscript. We hope the responses and revisions to our manuscript are thorough and precise. We look forward to hearing from you at your earliest convenience.

Sincerely Yours,

All the authors.

Reviewer(s)' Comments to Author:

Reviewer: 2

  1. The abstract is insufficiently described. Authors should include in this section the statistical analyzes carried out and specific -not generally- conclusions and/or practical implications.

Responses: Thanks for your critical comments. We have revised the abstract part as (See line 1-22):

Objectives: This study aimed to examine the associations between warm and harsh parenting and adolescent well-being and the mediating effects of self-kindness and self-judgment in relationships. Moreover, this study investigated different developmental across three adolescence stages (early, middle, and late).

Methods: Totally 14776 Chinese adolescents (mean age =13.53 ±2.08, 52.3% males), including early (10-12 years old, N=5055), middle (13-15 years old, N=6714), and late adolescence (16-18 years old, N=3007) participated in this study. All adolescents rated the levels of warm parenting and harsh parenting, self-kindness and self-judgment, and well-being. Structural equation modeling (SEM) was adopted to examine the mediation model. Multi-group analysis was conducted to investigate differences in the mediation model across different developmental stages.

Results: Both warm and harsh parenting were related to adolescent well-being through the mediating effects of self-kindness and self-judgment. But warm parenting exerted a more substantial impact on adolescent well-being. Self-kindness demonstrated a more robust mediating effect compared to self-judgment in relationships. Moreover, harsh parenting in late adolescence had a weaker impact on adolescent well-being than those in early and middle adolescence. Warm parenting had a more significant impact on adolescent well-being in early adolescence than in middle and late adolescence.

Conclusions: Overall, warm parenting had a more substantial effect on adolescent well-being than harsh parenting. The findings also highlighted the crucial mediating effect of self-kindness in the relationships between parenting and well-being. Moreover, this study also indicated the importance of warm parenting in early adolescence. Intervention programs should focus on enhancing the level of warm parenting to promote self-kindness in adolescents for improving well-being.

2.Authors could improve their literature review in the Introduction section about some aspects. First, I think it makes more sense and I recommend starting the introduction with a brief summary of the importance of the family and family patterns and how they are associated with adolescent adjustment. In the current paper, this summary is focused on the well-being variable. Second, the study is focused on three different adolescence stages (early, middle, and late). For this reason, the entire literature review should clearly reflect on with age range of adolescence the cited studies and not refer to the general adolescent population as the section is written right now. So, I recommend deleting the section entitled "developmental differences" and integrating this information throughout the introduction, since the entire study is carried out taking into account the different age ranges of this developmental period.

Responses: Thanks for your comments. Your suggestions are very critical and valuable for writing the introduction part. Hence, we added a brief summary of the importance of the family and family patterns and how they are associated with adolescent adjustment at the beginning of the introduction part (See line 47-80)

According to ecological systems theory (Bronfenbrenner, 1994; Darling, 2007; Ogg & Anthony, 2020), adolescent development is shaped by interactions between individuals and their contexts (e.g. family, school, community, society). The family has been acknowledged as the primary proximal socialization environment that exerts an immediate on adolescents (King et al., 2018; Moreira & Canavarro, 2018; Shek et al., 2021; Shek & Liang, 2018). Accordingly, adolescent adjustments are embedded in children’s proximal interaction processes with parents within the family (Bronfenbrenner, 1994; Ogg & Anthony, 2020), which has also been emphasized by the family system theory (Cox & Paley, 1997; Minuchin, 1985). Parenting behavior, constituting the most proximal family interaction environment (Hentges & Wang, 2018; Ogg & Anthony, 2020), has been shown to be important in promoting adolescent adjustments (Abidin et al., 2022; Hentges & Wang, 2018; Ren & Zhu, 2022; Zhu et al., 2021). The present study delves into the relationship between parenting behavior and adolescent well-being, with a particular emphasis on the significance of parenting style within the family setting. (See line 47-59)

According to the integrative model of parenting (Darling & Steinberg, 1993), parenting style,  referring to the emotional attitude of parents toward the child, plays a crucial role in shaping a good parent-child relationship and children’s adjustments (Abidin et al., 2022; Zhu & Shek, 2021). Parenting styles have been classically conceptualized as two broad parenting dimensions: responsiveness and demandingness (Baumrind, 1991). Responsiveness refers to parents providing support, responsive care, and sensitivity to children's needs to aid their development, while demandingness refers to supervising and enforcing discipline efforts (Baumrind, 1991; Zhang et al., 2017). The current study focused on warm and harsh parenting as the representative parenting styles for responsiveness and demandingness. Warm and harsh parenting are crucial in forming and maintaining a good parent-child relationship and have significant impacts on children’s multiple psychological adjustment outcomes (Chong et al., 2014; Etkin et al., 2014; Li et al., 2018; Liu & Wang, 2021; Medeiros et al., 2016). But few studies have simultaneously analyzed both warm and harsh parenting styles to determine their relative impact on adolescent well-being. Well-being refers to subjective feelings of happiness and enjoyment as well as healthy and prosperous individual functioning (King et al., 2018; Moreira & Canavarro, 2018). However, previous studies (Leung & Shek, 2019; Shek et al., 2021; Shek & Liang, 2018) assessing adolescent well-being have mainly used other indicators, such as life satisfaction and helplessness rather than adolescent well-being itself. Moreover, with particular reference to adolescents, their well-being is highly susceptible to external factors during the crucial developmental period (King et al., 2018; Moreira & Canavarro, 2018; Shek et al., 2021; Shek & Liang, 2018; Zhu & Shek, 2021). Hence, it is meaningful to investigate the relative importance of warm and harsh parenting for adolescent well-being. (See line 60-80)

  1. I recommend deleting figure 1. The mediation model should be well explained in the current study section and practically is the same as figure 2, it does not provide new information.

Responses: Thanks for your comments. Figure 1 was depicted to delineate our model construction and related hypothesis. Although it does not provide additional information, this Figure could help readers better capture and understand model construction and hypothesis formulation processes in the introduction part. Hence, after careful consideration, we choose to retain Figure 1 in our manuscript. Thanks for your suggestions again.

  1. The present study needs revision. Please, make sure all objectives and hypotheses are stated in this section and explain the results expected. Moreover, I recommend that authors delete all information related to study objectives throughout the introduction because they are repetitive and not friendly reading (e.g., “The current study investigated whether warm and harsh parenting, as two typical representations of parenting style dimensions (Baumrind, 1991; Zhang et al., 2017), could relate to adolescent well-being through the mediating effects of adolescent self-kindness and self-judgment” on page 5, “In this case, the current study examined the relationships between warm and harsh parenting and adolescent well-being” on page 6; “Therefore, the current study focused on self-kindness and self-judgment as potential mediators in relationships” on page 7).

Responses: Thanks for your comments. We have deleted the repetitive information related to study objectives throughout the introduction for clarity.

  1. I assume that based on the total sample (N = 14,776) the authors created the three groups of adolescents (early, middle, and late). This aspect should be clearly explained in the “Data analysis” section. For all measures used, please indicate whether response scores were summed or averaged to create their composite scores for data analysis.

Responses: Thanks for your comments. We have summed and standardized the composite score for data analysis. We added this information to the data analysis part as (See line 212-221):

Data analysis

Descriptive statistics and Pearson correlational analysis were conducted in SPSS 23.0. The present study performed the structural equation modeling (SEM) to examine the mediation model in Mplus 8.3. The bootstrap analysis with 2000 replicates and a 95% confidence interval was used to examine the significance of the mediation effects. Adolescent characteristics, including age, gender, and the only child status, were treated as control variables in the analysis. The standardized summed composite scores of each scale were used in the SEM analysis. Then we conducted a multi-group analysis to investigate differences in the mediation model across different developmental stages (Early adolescence:10-18 years old; Middle adolescence: 13-15 years old; Late adolescence: 16-18 years old).

  1. Because of parental patterns are different in function on cultural context, I believe the manuscript would gain value if in the Discussion section including a cultural lens to interpret the results found.

Response: Thanks for your important comments. We have added some discussions about the specific meanings of harsh parenting in a Chinese cultural context (See line 366-385)

In our study, we found that harsh parenting perceived by adolescents had a weaker impact on their well-being compared to warm parenting. This may be partially explained by the Chinese cultural context. Chinese culture, for example, attaches great importance to collectivism and filial piety, which is reflected in traditional values such as respect for elders, obedience to norms, and fulfillment of family obligations (Chao, 2000; Zhang et al., 2017). Compared to Western parents, Chinese parents are less likely to express love outwardly through physical affection or verbal affirmations (Deater-Deckard et al., 2011), more likely to adopt harsh discipline methods (Zhang et al., 2017), and attach less importance to autonomy (Supple et al., 2009). Additionally, Chinese parents often adhere to the traditional belief that they should be both strict and affectionate toward their children (Zhang et al., 2017). For example, a previous study (Kim et al., 2013) identified four types of parenting styles in Chinese American parents, including “supportive”, “easygoing”, “harsh,” and “tiger” parenting. Notably, a significant proportion of Chinese families were found to adopt harsh or “tiger” parenting (Kim et al., 2013). Under this circumstance, it is possible that adolescents in China may view strict or harsh parenting as a normal and expected part of family life. As a result, the negative effects of this type of parenting on adolescent well-being may be less pronounced. Furthermore, our study did not distinguish between fathers’ and mothers’ parenting roles. Considering the traditional image of parents, “strict father, kind mother” within families (Wang, 2017), parental gender may also play an important role in understanding the influence of harsh parenting on adolescent well-being. Future research may distinguish between fathers’ and mothers’ warm and harsh parenting to investigate the specific associations with adolescent well-being.

  1. Please describe the contributions of the study with more clarity and extensity.

Response: Thanks for your critical comments. We have added some discussion about the contributions of the study (See line 398-404).

Our findings contributed to understanding parenting and well-being in adolescents from the ecological perspective of development. By investigating the mediating mechanism of self-attitudes (self-kindness and self-judgment), our findings suggested that it important to improve self-kindness to promote adolescent well-being. Moreover, our study indicated the relative importance of warm parenting for adolescent well-being, particularly during early adolescence. Overall, our findings have indicated the significance of improving self-kindness in adolescents, particularly when adolescents experienced reduced warm and increased harsh parenting within families.

  1. At the end of the manuscript, the practical applications/implications of the study should be explained in detail.

Reponses: Thanks for your critical comments. We have added some discussion about the practical implications of the study at the end of the manuscript (See line 417-424).

Implications

Implications are also provided in this study. First, programs aimed at enhancing warm parenting and reducing harsh parenting may constitute important intervention directions to improve well-being among adolescents. Therefore, it may be particularly beneficial to improve warm parenting for adolescents. Second, it is suggested that self-kindness deserves a place to promote well-being among adolescents. Parents, teachers, and educators could facilitate the practice of self-kindness to help adolescent improve their well-being. Third, intervention programs should pay attention to improve self-kindness particularly during early adolescence.

Reviewer 3 Report

I congratulate the authors on their work. The article is interesting, original and important for the scientific community. 

I have only one suggestion, I think that Authors may write in the article's title a word that underlined the large number of parcipants, for example:

"Warm and harsh parenting, well being, self-kindness and self-judgment: An examination of developmental differences in a large sample of adolescents".

Kind regards

Author Response

Submission ID number: children-2140835

Title of paper: Warm and harsh parenting, self-kindness and self-judgment, well-being: An examination of developmental differences in a large sample of adolescents

Dear reviewers,

Thank you very much for your review of our manuscript. We greatly appreciate all your comments,

and these comments are helpful and constructive in revising this paper.

We have carefully revised the manuscript and addressed your comments. Our responses to each comment are enclosed in the response letter, and the revised parts have been highlighted in the revised manuscript. We hope the responses and revisions to our manuscript are thorough and precise. We look forward to hearing from you at your earliest convenience.

Sincerely Yours,

All the authors.

Reviewer(s)' Comments to Author:

Reviewer 3

I congratulate the authors on their work. The article is interesting, original and important for the scientific community. 

I have only one suggestion, I think that Authors may write in the article's title a word that underlined the large number of participants, for example:

"Warm and harsh parenting, well-being, self-kindness, and self-judgment: An examination of developmental differences in a large sample of adolescents".

Responses: Thanks for your critical comments. Your suggestion is very important for us. We have changed the title to “Warm and harsh parenting, self-kindness and self-judgment, well-being: An examination of developmental differences in a large sample of adolescents.

Best regards

Reviewer 4 Report

The article presents research aimed at checking the model of relationships between parenting style (measured in two dimensions) and the sense of well-being, self-kindness and self-judgment of adolescents in three stages of adolescence. A large group of adolescents was examined - during the pandemic (?) - so the question arises about the possible significance of the situation threatening the health and life of children - and parents, and the possible connection with the collected answers. Perhaps it would be good to at least note this fact in the text - it's about the time of data collection. It cannot be ruled out that repeating the research in a different ecological context (I mean the pandemic situation as well as the specificity of the culture in which the research was carried out) may bring different results or confirm those obtained by the authors. Both solutions (I mean the result of repeated research) would be valuable for understanding the importance of parental influences on children's development.

In connection with the results obtained by the authors of the article, a question arises: can it be said that parenting style (both warm and hard) is less important for the well-being of adolescents aged 16-18 than in earlier stages of development? This may be associated with the intensification of the need for autonomy from parents/attachment figures and focusing on peer relationships, as well as obtaining a higher level of self-trust (in the case of adolescents with good warm parenting style and secure attachment), or a more self-critical attitude (and perhaps even self-aggression) (in the case of "hard" parenting style).

I wonder if the hard parenting style manifests itself primarily in the use of corporal punishment against the child - this reflection results from the content of the items used in the study of the perception of the parental style by adolescents.

It is also worth considering the importance of the child's/adolescent's gender for the way of perceiving/evaluating the parental style and for their attitude towards themselves (self-kindness and self-judgment).

I propose to consider recalling (at least signaling) U. Bronfenbrenner's views contained in the ecological concept of development (Bronfenbrenner's Ecological Systems Theory) (eg 1977).

The article is interesting and may be the basis for undertaking new explorations concerning other manifestations of youth development, self-concept, behavior, or generally: adaptation to the context of life and forecasts of further development.

Author Response

Submission ID number: children-2140835

Title of paper: Warm and harsh parenting, self-kindness and self-judgment, well-being: An examination of developmental differences in a large sample of adolescents

Dear reviewers,

Thank you very much for your review of our manuscript. We greatly appreciate all your comments,

and these comments are helpful and constructive in revising this paper.

We have carefully revised the manuscript and addressed your comments. Our responses to each comment are enclosed in the response letter, and the revised parts have been highlighted in the revised manuscript. We hope the responses and revisions to our manuscript are thorough and precise. We look forward to hearing from you at your earliest convenience.

Sincerely Yours,

All the authors.

Reviewer(s)' Comments to Author:

Reviewer 4

1、The article presents research aimed at checking the model of relationships between parenting style (measured in two dimensions) and the sense of well-being, self-kindness and self-judgment of adolescents in three stages of adolescence. A large group of adolescents was examined - during the pandemic (?) - so the question arises about the possible significance of the situation threatening the health and life of children - and parents, and the possible connection with the collected answers. Perhaps it would be good to at least note this fact in the text - it's about the time of data collection. It cannot be ruled out that repeating the research in a different ecological context (I mean the pandemic situation as well as the specificity of the culture in which the research was carried out) may bring different results or confirm those obtained by the authors. Both solutions (I mean the result of repeated research) would be valuable for understanding the importance of parental influences on children's development.

Responses: Thanks for your important comments. This study was conducted during December 2021, which was still during the pandemic. Hence, we have added some discussion about the possible influence of the situation threatening family life and the collected answers. As you suggested, repeating the research in a different ecological context (e.g., after the pandemic or in Western culture) may bring different results. Hence, we added some discussion about the time of data collection (during the pandemic) and the cultural context (Chinese culture) in interpreting the results (See line 366-396).

In our study, we found that harsh parenting perceived by adolescents had a weaker impact on their well-being compared to warm parenting. This may be partially explained by the Chinese cultural context. Chinese culture, for example, attaches great importance to collectivism and filial piety, which is reflected in traditional values such as respect for elders, obedience to norms, and fulfillment of family obligations (Chao, 2000; Zhang et al., 2017). Compared to Western parents, Chinese parents are less likely to express love outwardly through physical affection or verbal affirmations (Deater-Deckard et al., 2011), more likely to adopt harsh discipline methods (Zhang et al., 2017), and attach less importance to autonomy (Supple et al., 2009). Additionally, Chinese parents often adhere to the traditional belief that they should be both strict and affectionate toward their children (Zhang et al., 2017). For example, a previous study (Kim et al., 2013) identified four types of parenting styles in Chinese American parents, including “supportive”, “easygoing”, “harsh,” and “tiger” parenting. Notably, a significant proportion of Chinese families were found to adopt harsh or “tiger” parenting (Kim et al., 2013). Under this circumstance, it is possible that adolescents in China may view strict or harsh parenting as a normal and expected part of family life. As a result, the negative effects of this type of parenting on adolescent well-being may be less pronounced. Furthermore, our study did not distinguish between fathers’ and mothers’ parenting roles. Considering the traditional image of parents, “strict father, kind mother” within families (Wang, 2017), parental gender may also play an important role in understanding the influence of harsh parenting on adolescent well-being. Future research may distinguish between fathers’ and mothers’ warm and harsh parenting to investigate the specific associations with adolescent well-being.

Finally, this study was conducted during the pandemic in China. Empirical studies have indicated the adverse influence of the pandemic on families (Feinberg et al., 2022; Overall et al., 2022; Spinelli et al., 2021). During the pandemic, it is pretty stressful for parents to balance multiple demands from work, life, and family (Zou et al., 2022). Parents experienced elevated levels of parenting stress (Spinelli et al., 2021), which may increase the possibility of adopting more harsh parenting (Chung et al., 2020). Additionally, a recent study (Feinberg et al., 2022) reported a significant deterioration in parenting quality during the pandemic’s initial months compared to pre-pandemic. Hence, researchers (Feinberg et al., 2022) stressed the importance of intervention programs to prevent prolonged individual and family problems by offering more support and guidance to families.

2、In connection with the results obtained by the authors of the article, a question arises: can it be said that parenting style (both warm and hard) is less important for the well-being of adolescents aged 16-18 than in earlier stages of development? This may be associated with the intensification of the need for autonomy from parents/attachment figures and focusing on peer relationships, as well as obtaining a higher level of self-trust (in the case of adolescents with good warm parenting style and secure attachment), or a more self-critical attitude (and perhaps even self-aggression) (in the case of "hard" parenting style).

Responses: Thanks for your valuable comments. We have added some discussion about the differences across adolescence stages (See line 341-349)

In terms of the results, we found that perceived parenting style exerted stronger impacts on adolescent well-being during early adolescence relative to late adolescence stage. The results were similar to previous findings (Shek et al., 2021; Zhang et al., 2015) indicating different patterns of parenting influence across adolescence stages. For example, parental psychological control have different effects on meaning of life in early and late adolescence stages (Shek et al., 2021), possibly due to the growing autonomy and independence and reduced susceptibility to parenting. From another point, this may also be partially explained by the developmental changes of parenting (Lansford et al., 2021), such as decreases in warmth and behavioral control as children age from 8 to 16, with shaper declines particularly after early adolescence.

3、I wonder if the hard parenting style manifests itself primarily in the use of corporal punishment against the child - this reflection results from the content of the items used in the study of the perception of the parental style by adolescents.

Responses: Thanks for your valuable comments. In the current study, we adopted the brief version of the Alabama Parenting Questionnaire (APQ) (Lu et al., 2019), which measured harsh parenting using items referring to the use of corporal punishment against the child. Harsh parenting involves negative, punitive, and dysregulated emotional reactions directed at children, such as yelling, hitting, and coercion (Hentges & Wang, 2018), which have been confirmed to be associated with multiple adjustments outcomes of adolescents (Hentges & Wang, 2018; Martoccio et al., 2016; McKinney & Szkody, 2018). Even though the definitions may vary in studies, it is usually considered a method to regulate or control the children's behavior by adopting physical or psychological force to inflict physical or emotional harm (Straus et al., 1998; Wang & Liu, 2018), Psychological aggression and corporal punishment were the predominant forms of harsh parenting in both Western and Chinese societies (Liu et al., 2022; Wang & Liu, 2014, 2018). Hence, we utilized the items of corporal punishment to represent harsh parenting to investigate its influence on adolescent well-being.  

4、It is also worth considering the importance of the child's/adolescent's gender for the way of perceiving/evaluating the parental style and for their attitude towards themselves (self-kindness and self-judgment).

Responses: Thanks for your valuable comments. We have added some discussion about possible gender differences in the Future directions part (See line 351-365).

This study did not examine possible gender differences. Previous studies have suggested that child gender would influence the way of perceiving parenting style, its impacts on psychological adjustments (Wang et al., 2023; Wang, 2017; Zhu & Shek, 2021), and on their self-attitudes (Bluth et al., 2017; Liu et al., 2020; Yarnell et al., 2018). For example, one study (Zhu & Shek, 2021) found a stronger association between parental psychological control and adolescent well-being in adolescent girls than boys. Particularly for girls, harsh parenting would adversely influence adolescents’ meaning in life and self-control, which enhances their problematic smartphone use, (Wang et al., 2023). Additionally, harsh parenting was only directly and indirectly related to female depression, but not male depression (McKinney & Szkody, 2018), highlighting the specific importance of harsh parenting for girls. Researchers (Bluth et al., 2017; Liu et al., 2020; Yarnell et al., 2018) have also revealed potential gender differences in terms of self-attitudes. For example, self-identified men tended to have more self-compassion toward themselves than women (Yarnell et al., 2018). Older females had the lowest self-compassion levels compared to younger females or all-age males (Bluth et al., 2017). Future research may adopt multiple methods and measures to examine possible age and gender differences in these associations.

5、I propose to consider recalling (at least signaling) U. Bronfenbrenner's views contained in the ecological concept of development (Bronfenbrenner's Ecological Systems Theory) (eg 1977).

Responses: Thanks for your critical comments. We have added Bronfenbrenner's views contained in the ecological concept of development to the Introduction part. (See line 47-59)

According to ecological systems theory (Bronfenbrenner, 1994; Darling, 2007; Ogg & Anthony, 2020), adolescent development is shaped by interactions between individuals and their contexts (e.g. family, school, community, society). The family has been acknowledged as the primary proximal socialization environment that exerts an immediate on adolescents (King et al., 2018; Moreira & Canavarro, 2018; Shek et al., 2021; Shek & Liang, 2018). Accordingly, adolescent adjustments are embedded in children’s proximal interaction processes with parents within the family (Bronfenbrenner, 1994; Ogg & Anthony, 2020), which has also been emphasized by the family system theory (Cox & Paley, 1997; Minuchin, 1985). Parenting behavior, constituting the most proximal family interaction environment (Hentges & Wang, 2018; Ogg & Anthony, 2020), has been shown to be important in promoting adolescent adjustments (Abidin et al., 2022; Hentges & Wang, 2018; Ren & Zhu, 2022; Zhu et al., 2021). The present study delves into the relationship between parenting behavior and adolescent well-being, with a particular emphasis on the significance of parenting style within the family setting.

6、The article is interesting and may be the basis for undertaking new explorations concerning other manifestations of youth development, self-concept, behavior, or general: adaptation to the context of life and forecasts of further development.

Reponses: Thanks for your comments and suggestions very much!

Round 2

Reviewer 2 Report

The authors were very responsive with regard all my observations. Special thanks to the authors for introducing the cultural lens in the discussion of the manuscript, I think the manuscript has gained in quality. I believe that now the manuscript is ready to be accepted.